# Moiré-like Superlattice Generated van Hove Singularities in a Strained CuO$_2$ Double Layer

**Artem O. Sboychakov** [1,†] , **Kliment I. Kugel** [1,2,*,†] **and Antonio Bianconi** [3,4,5,†]

1    Institute for Theoretical and Applied Electrodynamics, Russian Academy of Sciences, 125412 Moscow, Russia
2    Institute of Metal Physics, Ural Branch, Russian Academy of Sciences, 620990 Ekaterinburg, Russia
3    Rome International Center for Materials Science Superstripes RICMASS, Via dei Sabelli 119A, 00185 Rome, Italy
4    Institute of Crystallography, CNR, Via Salaria Km 29.300, 00015 Rome, Italy
5    Moscow Engineering Physics Institute, National Research Nuclear University MEPhI, Kashirskoe sh. 31, 115409 Moscow, Russia
\*    Correspondence: klimkugel@gmail.com
†    These authors contributed equally to this work.

**Abstract:** While it is known that the double-layer Bi$_2$Sr$_2$CaCu$_2$O$_{8+y}$ (BSCCO) cuprate superconductor exhibits a one-dimensional (1D) incommensurate superlattice (IS), the effect of IS on the electronic structure remains elusive. Following the recent shift of interest from an underdoped phase to optimum and overdoped phases in BSCCO by increasing the hole doping $x$, controlled by the oxygen interstitials concentration $y$, here we focus on the multiple splitting of the density of states (DOS) peaks and emergence of higher order van Hove singularities (VHS) due to the 1D incommensurate superlattice. It is known that the 1D incommensurate wave vector $\mathbf{q} = \epsilon \mathbf{b}$ (where $\mathbf{b}$ is the reciprocal lattice vector of the orthorhombic lattice) is controlled by the misfit strain between different atomic layers in the range 0.209–0.215 in BSCCO and in the range 0.209–0.25 in Bi$_2$Sr$_2$Ca$_{1-x}$Y$_x$Cu$_2$O$_{8+y}$ (BSCYCO). This work reports the theoretical calculation of a complex pattern of VHS due to the 1D incommensurate superlattice with large 1D quasi-commensurate supercells with the wave vector $\epsilon = 9/\eta$ in the range $36 > \eta > 43$. The similarity of the complex VHS splitting and appearing of higher order VHS in a mismatched CuO$_2$ bilayer with VHS due to the moiré lattice in strained twisted bilayer graphene is discussed. This makes a mismatched CuO$_2$ bilayer quite promising for constructing quantum devices with tuned physical characteristics.

**Keywords:** cuprate bilayers; superstructures; band structure; electron density of states; van Hove singularities

## 1. Introduction

Different van der Waals heterostructures [1], such as bilayer graphene [2] and transition metal dichalcogenide bilayers [3,4], provide an actively progressing testing ground for a variety of quantum phenomena, e.g., flat bands and electron nematicity. Indeed, if we put together two layers of van der Waals materials, a lattice mismatch or twist angle leads to the formation of moiré patterns with large supercells. This provides a promising means of the band and crystal structure engineering, leading, in particular, to the moiré modulated topological order [5], and paving way to such novel fields of research as twistronics [6].

Bilayers of atomic cuprate oxide superconductors appear to be not less interesting. The lattice parameter and aperiodic local lattice distortions associated with variable microstrain are not key physical parameters in BCS or unconventional superconductivity theories. Nevertheless, compelling evidence for the key roles of the CuO$_2$ lattice in the mechanism of high-temperature superconductivity and charge density wave (CDW) phenomena in cuprate perovskites has been reported [7–11]. These findings confirm early experimental results provided by local and fast experimental methods based on the use of

synchrotron X-ray radiation [12–16] pointing toward the relevant role of the micro-strain in superconducting cuprate perovskites [17,18].

The $Bi_2Sr_2CaCu_2O_{8+y}$ (BSCCO) crystal [14–16] is a misfit layer compound [19] exhibiting an incommensurate composite structure [19,20] and large atomic displacements from average positions in the [BiO], [SrO], [CuO$_2$], and [Ca] layers. These displacements form an incommensurate composite structure, where nanoscale insulating and metallic atomic stripes—see, for example, [21–24]—of $Bi_2Sr_2Ca_{1-x}Y_xCu_2O_{8+y}$ (BSCYCO) with six different elements in the average crystalline unit cell form an archetypal case of high entropy perovskites (HEPs). The search for new HEPs is today a hot topic focusing on the manipulation of the complexity, where the entropy control plays a key role in the determination of material functionalities, as was observed in bismuth 2212 iron ferrite and ferrate compounds [25,26], thermoelectric misfit layered cobaltite oxide perovskites [27,28], and in the form of 2D flakes [29], by taking advantage of the polymorphisms of perovskite's structure with competing nanoscale crystallographic phases [30,31].

Spatially correlated incommensurate lattice stripy modulations in $Bi_2Sr_2CaCu_2O_{8+y}$ superconductors can now be visualized by the scanning submicron X-ray diffraction, focusing on the superlattice satellites of the main X-ray reflections [32–34]. The results show that, while oxygen concentration and the doping level can change depending on the sample treatments and thermal history, the incommensurate superlattice wave vector is controlled by the misfit strain, and it exhibits remarkable stability. The critical temperature exhibits a dome as a function of oxygen concentration or by potassium deposition [35], but the maximum critical temperature of the dome [36] is controlled by the lattice stripe aperiodic structure [37] and misfit strain [17].

Recently, the scientific interest turned toward the normal state of cuprate perovskites exhibiting high-temperature superconductivity in the overdoped regime in the proximity of the superconductor-to-metal transition tuned by doping, and for many years the focus was on the underdoped regime. The investigations of the overdoped phase [38–45] of Bi2212 have found evidence that the superconducting gap appears at temperatures well above the critical temperature $T_c$, whereas below $T_c$, a large fraction of the normal state remains uncondensed. Scanning tunneling microscopy has shown that on the edge of the overdoped regime, the gap remains in isolated puddles above $T_c$, so that superconducting puddles are intercalated by a normal metal phase. Moreover, at the same time, the Fermi surface topology changes, giving rise to an electronic topological Lifshitz transition. This provides compelling evidence for a nanoscale phase separation, giving a granular landscape, where superconducting puddles are embedded in a normal phase. The phase separation scenario was predicted to appear at the Lifshitz transition several years ago [46–50].

It is known that the interlayer interaction between $CuO_2$ of BSCCO planes produces a bilayer splitting of the 2D $Cu(3d)O(2p)$ single band into two bands with even and odd symmetry exhibiting a double-peaked logarithm divergent density of states (DOS) due to standard van Hove singularities (VHS) in a 2D metal. However, it is not known how the 1D superstructure controls the VHS.

While many electronic band structure calculations of the average structure BSCCO have been published in these last 33 years (see, e.g., the papers beginning from [51] up to one of the most recent papers [52]), to our knowledge, the electronic band structure calculations of BSCCO with the incommensurate superstructure with the wave vector $0.209 < q < 0.215$ (see e.g., [53]) known since 1989, has never been reported. Here, we report the band structure calculation of the strained $CuO_2$ bilayer with the incommensurate structure, focusing on the high doping range, where correlations become negligible and the van Hove singularity crosses the chemical potential. We focus on the variation of the electronic topological transition from a hole Fermi surface to an electron Fermi surface in the overdoped regime in the BSCCO double layer perovskite, while taking into account the aperiodic incommensurate lattice modulation.

In $Bi_2Sr_2CaCu_2O_{8+y}$, the mobility of oxygen interstitials in the $CuO_2$ planes is high in the temperature range 200–380 K, and it is possible to control the concentration $y$ of oxygen interstitials to change the hole doping $x$ in the layered perovskite.

The insulating $Bi_2Sr_2YCu_2O_8$ exhibits a commensurate modulation with $\varepsilon = 0.25$, which indicates the formation of lattice stripes with commensurate period $n = 1/\varepsilon = 4$ lattice units. The superconducting $Bi_2Sr_2CaCu_2O_{8+y}$ exhibits a commensurate modulation with $\varepsilon = 0.209$, which indicates the formation of a lattice stripes superlattice with incommensurate period $n = 1/0.209 = 43/9$ lattice units. For $\varepsilon = 0.2$, the $CuO_2$ lattice should be decorated by a commensurate-stripes superlattice with $n = 1/\varepsilon = 1/0.2 = 5$ lattice units. For intermediate values of the superlattice wave vector $0.25 < \varepsilon < 0.2$, the incommensurate superstructure [54] consists of the mixture of the $\varepsilon = 1/n$ and $\varepsilon = 1/m$ order, i.e., alternating the so called $n = 4$ stripes portions and $m = 5$ stripes portions. Indeed, quasi-commensurate periods $\lambda = 1/\varepsilon = (nx + my)/(n + m)$, intermediate between two main commensurate wave vectors $1/n$ and $1/m$ with periods of $m$ and $n$ lattice units, respectively, occur at integer numbers of lattice unit cells $\eta = (n + m)/\varepsilon$. Each quasi-commensurate phase (QCP) corresponds to a modulation wave locked with the underlying lattice onto a rational number. The sequence of quasi-commensurate phases is called the devil's staircase [54], which has been observed in many complex materials.

The quasi-commensurate modulation period in BSCCO is given by $1/\varepsilon = (4x + 5y)/(4 + 5)$, where $x + y = 9$. Therefore, $\eta = 9/\varepsilon = 4x + 5y$ varies in the range $36 < \eta < 45$, and at each integer value of $\eta$, the system reaches a quasi-commensurate phase at each step of a *devil's staircase*. We have found that in the BSCCO samples, where the chemical potential is tuned by the concentration of oxygen interstitials, the mismatched $CuO_2$ bilayer is tuned by the misfit strain at the quasi-commensurate modulation with $\eta = 43$, i.e, $\epsilon=9/43 = 0.209$ in agreement with the recent scanning micro-X-ray diffraction [34], which is the *devil's staircase* for $x = 2$ and $y = 5$. Therefore, we have $4 \times 2 + 5 \times 7 = 43$ (i.e., a quasicrystal made of eight insulating lattice units and 35 metallic lattice units).

## 2. Geometry of the System

We modeled the system under study as consisting of two mismatched $CuO_2$ layers. Layer 1 is assumed to be quadratic. For the reason described below, we chose the reference frame as shown on the left panel in Figure 1. For such a choice, the vectors connecting next nearest-neighbor Cu ions have coordinates $\mathbf{d}_1 = d(1, 0)$ and $\mathbf{d}_2 = d(0, 1)$, where $d$ is the distance between nearest copper ions in the *diagonal* direction. We chose [14] $d = 5.37\,\text{Å}$. We also introduced vectors connecting nearest-neighbor Cu ions of layer 1 (lattice vectors of the unit cell of layer 1). They are expressed via vectors $\mathbf{d}_{1,2}$ according to $\mathbf{a}_1 = (\mathbf{d}_1 - \mathbf{d}_2)/2$ and $\mathbf{a}_2 = (\mathbf{d}_1 + \mathbf{d}_2)/2$. The positions of Cu ions in the layer 1 are

$$\mathbf{r}_{1\mathbf{n}}^d \equiv \mathbf{r}_{1\mathbf{n}} = n\mathbf{a}_1 + m\mathbf{a}_2\,, \tag{1}$$

where $\mathbf{n} = (n, m)$ ($n, m$ are integers), and the positions of oxygen ions are

$$\mathbf{r}_{1\mathbf{n}}^{p_x} = \mathbf{r}_{1\mathbf{n}} + \frac{1}{2}\mathbf{a}_1\,, \quad \mathbf{r}_{1\mathbf{n}}^{p_y} = \mathbf{r}_{1\mathbf{n}} + \frac{1}{2}\mathbf{a}_2\,. \tag{2}$$

Layer 2 is assumed to be stretched or compressed in the $x$ (diagonal) direction. As a result, the vectors connecting next nearest-neighbor Cu ions in this layer have coordinates $\mathbf{d}_1' = d(1 - \delta, 0)$ and $\mathbf{d}_2' = d(0, 1)$, where parameter $\delta$ describes the strength of the stretching ($\delta < 0$) or compression ($\delta > 0$). The vectors connecting nearest-neighbor Cu ions of the layer 2 become $\mathbf{a}_1' = (\mathbf{d}_1' - \mathbf{d}_2')/2$ and $\mathbf{a}_2' = (\mathbf{d}_1' + \mathbf{d}_2')/2$. The positions of copper and oxygen ions in the layer 2 are

$$\begin{aligned}
\mathbf{r}_{2\mathbf{n}}^d &\equiv \mathbf{r}_{2\mathbf{n}} = n\mathbf{a}_1' + m\mathbf{a}_2'\,, \\
\mathbf{r}_{2\mathbf{n}}^{p_x} &= \mathbf{r}_{2\mathbf{n}} + \frac{1}{2}\mathbf{a}_1'\,, \quad \mathbf{r}_{2\mathbf{n}}^{p_y} = \mathbf{r}_{2\mathbf{n}} + \frac{1}{2}\mathbf{a}_2'\,.
\end{aligned} \tag{3}$$

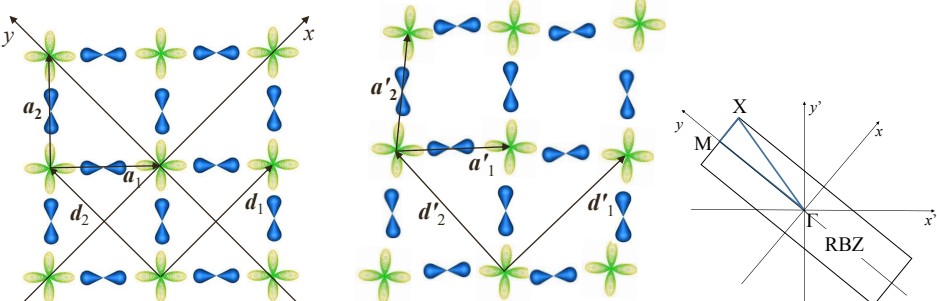

**Figure 1.** Schematics of the $CuO_2$ layers 1 (**left** panel) and 2 (**middle** panel). Layer 1 is unstretched. Layer 2 is either stretched of compressed in the diagonal ($x$) direction. As a result, for some commensurate stretching/compression, a superstructure appears. The superlattice contains $2N_1$ elementary unit cells of the layer 1 and $2N_2$ elementary unit cells of layer 2. The right figure shows the sketch of the reduced Brillouin zone of the superlattice. The $\Gamma$ point is in the coordinate origin, the $M$ point has coordinates $\mathbf{M} = (0, \pi/d)$, and the coordinate of the $X$ point is $\mathbf{X} = (\pi/(N_1 d), \pi/d)$. The thick blue triangle shows the contour, along which the spectra (see below) are calculated.

For a rational value of $1 - \delta = N_1/N_2$ ($N_1$ and $N_2$ are co-prime positive integers), the system has a superstructure. To satisfy the periodicity conditions, we must include two adjacent chains of copper and oxygen ions into superlattice cell aligned in the diagonal ($x$) direction. As a result, the superlattice cell will contain $2N_1$ copper ions of layer 1 and $2N_2$ copper ions of layer 2. It also will contain $4N_1$ oxygen ions of layer 1 and $4N_2$ oxygen ions of layer 2. The superlattice vectors are $\mathbf{R}_1 = N_1 \mathbf{d}_1 = N_2(1 - \delta)\mathbf{d}_1$ and $\mathbf{R}_2 = 2\mathbf{d}_2$. The superlattice Brillouin zone (reduced Brillouin zone, RBZ) has a shape of a rectangle with the dimensions $2\pi/(N_1 d)$ in the $x$ direction and $2\pi/d$ the $y$ direction (see the right panel of Figure 1).

### 3. Tight-Binding Model of Mismatched $CuO_2$ Bilayer

To describe electronic properties of mismatched $CuO_2$ bilayer, we use the four-band tight-binding model. The model includes two types of $p$ oxygen orbitals: $p_x$ orbitals of oxygen ions located in positions $\mathbf{r}_{i\mathbf{n}}^{p_x}$ and $p_y$ orbitals of oxygen ions located in positions $\mathbf{r}_{i\mathbf{n}}^{p_y}$. Model also includes two types of $d$ orbitals of copper located in positions $\mathbf{r}_{i\mathbf{n}}^{d}$: these are $x^2 - y^2$ and "axial" ($a$) orbitals. We do not specify here the nature of the latter $a$ orbital, and only mention its circular symmetry in $xy$ plane. It can be, e.g., $s$ or $3z^2 - r^2$ orbital of copper, or their superposition. We include this orbital in the model because it gives the largest effect on the interlayer hybridization [55].

The Hamiltonian under study can be written as

$$H = H_1 + H_2 + H_{12}. \tag{4}$$

The terms $H_{1,2}$ correspond to each individual layer, and $H_{12}$ describes the interlayer hybridization. Let us consider first intralayer terms. We used the simplest intralayer tight-binding Hamiltonian including only nearest neighbor hoppings of electrons between nearest copper and oxygen ions. The Hamiltonian $H_i$ ($i = 1, 2$) reads

$$H_i = \sum_{\mathbf{n}A\sigma} \varepsilon_A d_{\mathbf{n}iA\sigma}^\dagger d_{\mathbf{n}iA\sigma} + \sum_{\substack{\langle \mathbf{nm} \rangle \\ dp\sigma}} \left( t_{i\mathbf{nm}}^{dp} d_{\mathbf{n}id\sigma}^\dagger d_{\mathbf{m}ip\sigma} + H.c. \right), \tag{5}$$

where the subscript $A = \{d, p\}$ (with $d = x^2 - y^2$, $a$ and $p = p_x$, $p_y$) denotes the considered orbitals, and $d_{\mathbf{n}iA\sigma}^\dagger$ and $d_{\mathbf{n}iA\sigma}$ are the creation and annihilation operators of an electron located in unit cell $\mathbf{n}$ of layer $i$ having orbital index $A$ and spin projection $\sigma$. The first term in (5) includes the local energies of electrons in different orbitals, and the last term describes the nearest-neighbor hopping. The hopping amplitudes $t_{i\mathbf{nm}}^{dp}$ are position dependent. For

a given direction of hopping $\mathbf{m} \to \mathbf{n}$, the hopping amplitudes $t_{i\mathbf{nm}}^{dp}$ can we written in the form of a $2 \times 2$ matrix. The relationships between elements of this matrix can be easily obtained from the symmetry analysis of orbitals' wave functions [56]. Let us introduce the Fourier transformed electron operators $d_{\mathbf{k}iA\sigma}$ and Fourier transformed hopping amplitudes $t_{i\mathbf{k}}^{dp}$. Taking into account the symmetry of the orbitals under study [56], one can write for layer 1

$$t_{1\mathbf{k}}^{dp} = \begin{pmatrix} -t_{1p}e^{-i\mathbf{k}\mathbf{a}_1/2} & t_{1p}e^{-i\mathbf{k}\mathbf{a}_2/2} \\ -t_{2p}e^{-i\mathbf{k}\mathbf{a}_1/2} & -t_{2p}e^{-i\mathbf{k}\mathbf{a}_2/2} \end{pmatrix}, \tag{6}$$

where we introduce two independent Slater–Koster parameters describing the hopping between $x^2 - y^2$ and $p$ orbitals ($t_{1p}$) and between $a$ and $p$ orbitals ($t_{2p}$). The similar formula for layer 2 can be obtained from Equation (6) with the change $\mathbf{a}_{1,2} \to \mathbf{a}'_{1,2}$. Strictly speaking, the values of parameters $t_{1p}$ and $t_{2p}$ for layer 2 are different from those for layer 1. In further analysis, however, we neglect this difference due to the smallness of the lattice distortion. Following [55], we used the following values of the tight-binding parameters: $\varepsilon_{x^2-y^2} = 0$, $\varepsilon_{p_x} = \varepsilon_{p_y} = -0.9\,\mathrm{eV}$, $t_{1p} = 1.6\,\mathrm{eV}$ and $t_{2p} = 1.5\,\mathrm{eV}$. In our simulations, we used different values of the parameter $\varepsilon_a$ describing the on-site energy of $a$ orbital, taking both negative (almost $3z^2 - r^2$ orbital) and positive (almost $s$ orbital) values. We also performed simulations for different values of the parameter $t_{2p}$ describing hybridization between $a$ orbitals of copper and $p$ oxygen orbitals. Independent of the values of $\varepsilon_a$ and $t_{2p}$, the density of states of individual layer has only one van Hove peak corresponding to the change in the Fermi surface topology. The position of this peak depends on $\varepsilon_a$ and $t_{2p}$. In the present study, we always focus on the energy range close to the van Hove peak. The obtained results are similar for different values of $\varepsilon_a$ and $t_{2p}$.

Let us consider now the interlayer hopping. According to [55], the largest interlayer hopping amplitudes are between $a$ orbitals in two different layers, and between $a$ and $p_{x,y}$ orbitals of different layers. As a result, Hamiltonian $H_{12}$ takes the form

$$\begin{aligned} H_{12} &= \sum_{\mathbf{nm}p\sigma}\left( t_{\perp\mathbf{nm}}^{ap} d_{\mathbf{n}2a\sigma}^{\dagger} d_{\mathbf{m}1p\sigma} + t_{\perp\mathbf{nm}}^{pa} d_{\mathbf{n}2p\sigma}^{\dagger} d_{\mathbf{m}1a\sigma}\right) + \\ & \sum_{\mathbf{nm}\sigma} t_{\perp\mathbf{nm}}^{aa} d_{\mathbf{n}2a\sigma}^{\dagger} d_{\mathbf{m}1a\sigma} + H.c. \end{aligned} \tag{7}$$

The value of hopping amplitude depends on the mutual positions of two orbitals. In our simulations, we tried different parametrizations of interlayer hopping amplitudes depending on the possible symmetry of $a$ orbital and obtained qualitatively similar results. If an $a$ orbital is assumed to be of $s$ symmetry, the amplitude $t_{\perp\mathbf{nm}}^{ap_{x,y}}$ of electron hopping from the oxygen ion in layer 1 located in the position $\mathbf{r}_{1\mathbf{m}}^{p_{x,y}}$ to the copper ion in layer 2 located in the position $\mathbf{r}_{2\mathbf{n}}^{d}$ is described by the following Slater–Koster formula:

$$t_{\perp\mathbf{nm}}^{ap_{x,y}} = \frac{(\mathbf{r}_{2\mathbf{n}}^{d} - \mathbf{r}_{1\mathbf{m}}^{p_{x,y}})\mathbf{e}_{x,y}}{\sqrt{c^2 + (\mathbf{r}_{2\mathbf{n}}^{d} - \mathbf{r}_{1\mathbf{m}}^{p_{x,y}})^2}} V_\sigma(\mathbf{r}_{2\mathbf{n}}^{d} - \mathbf{r}_{1\mathbf{m}}^{p_{x,y}}), \tag{8}$$

where $\mathbf{e}_{x,y}$ is the unit vector in the direction of $\mathbf{a}_{1,2}$ and the function

$$V_\sigma(r) = t_0 \sqrt{1 + 8c^2/d^2}\, e^{-\frac{r - \sqrt{c^2 + d^2/8}}{r_0}}. \tag{9}$$

In the expression above, $t_0$ is the largest interlayer hopping amplitude between $a$ and $p$ orbitals and $c$ is the interlayer distance. We chose [14] $c = 3.35\,\text{Å}$ (note that the unit cell of the bulk BSCCO contains eight CuO$_2$ layers). The parameter $r_0$ defines how fast the function $V_\sigma(r)$ decays with the distance between orbitals (we chose $r_0/d = 0.19$).

Following [55], we took $t_0 = 0.27$ eV. For the hopping amplitude between $a$ orbitals in different layers, we have

$$t^{aa}_{\perp \mathbf{nm}} = V_0(\mathbf{r}^d_{2\mathbf{n}} - \mathbf{r}^d_{1\mathbf{m}}), \quad V_0(\mathbf{r}) = t^{aa}_0 e^{-\frac{r-c}{r^{aa}_0}}, \tag{10}$$

where we chose $t^{aa}_0 = 0.75$ eV and $r^{aa}_0 / d = 0.3$.

Following the Slater–Koster formalism [56], we tried also a somewhat complicated parametrization of interlayer hopping amplitudes, which corresponds to an $a$ orbital rather of $3z^2 - r^2$ symmetry. Again, the functions, similarly to $V_\sigma(r)$ and $V_0(r)$, contain factors depending on directional cosines and the factors describing exponential decay with the distance between ions, but now the hopping amplitudes contain largers number of such functions. Details of this parametrization can be found elsewhere. Here, we present the results corresponding to the first parametrization, but let us notice again that qualitatively the results are independent of the type of the hopping amplitude parametrization.

## 4. Results

Here, we demonstrate the evolution of the density of states and energy spectra for bilayers at different values of the lattice mismatch. We consider the superstructures with fixed $N_1 = 36$ and different $N_2$ ranging from $N_2 = 36$ (no mismatch) to $N_2 = 45$ (compressive strain). We suggest that apical orbital $a$ is rather the $s$ orbital of copper, so its energy level lies above the $x^2 - y^2$ orbital, and $\varepsilon_a = 6.5$ eV. In this case, the undoped bilayer has five electrons inside an elementary unit cell of each layer (four electrons on $p_{x,y}$ orbitals plus one electron on $d_{x^2-y^2}$ orbital). We are interested in the hole doped bilayers with doping $0 < x = 5 - n < 1$, where $n$ is the number of electrons per $CuO_2$ unit cell. Let us note here that $x$ differs from the oxygen content $y$ in the chemical formula of $Bi_2Sr_2CaCu_2O_{8+y}$. For example, according to [33], we have $x = y - 0.05$ for some $y$. In the interesting energy range, the density of states of the individual layer has one van Hove singularity. For unmismatched layers, the interlayer hopping terms split this van Hove singularity into two. Their energy positions depend on the model parameters. Thus, increasing the hopping amplitude between $a$ and $p_{x,y}$ orbitals, $t_{2p}$, shifts these peaks to larger energies (smaller doping) in agreement with [57]. For example, in Figure 2, we show the densities of states at the Fermi level as functions of doping $x$ calculated for three different values of $t_{2p}$. Below, we consider the case of $t_{2p} = 1.5$ eV as more realistic from the viewpoint of the positions of van Hove singularities in real compounds.

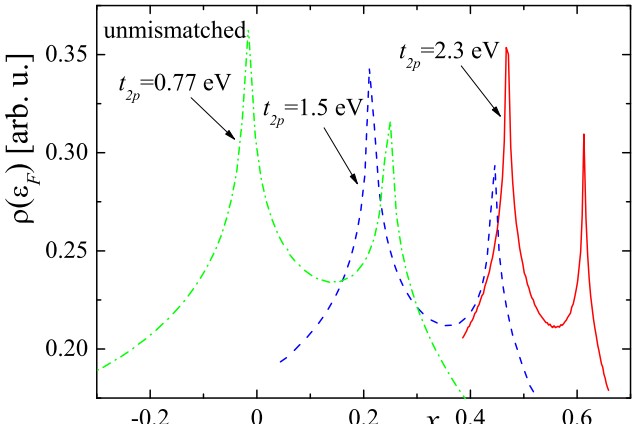

**Figure 2.** Densities of states at the Fermi level for unmismatched layers as functions of doping $x$, calculated for three different values of $t_{2p}$. Other model parameters are: $\varepsilon_{x^2-y^2} = 0$, $\varepsilon_{p_x} = \varepsilon_{p_y} = -0.9$ eV, $\varepsilon_a = 6.5$ eV, $t_{1p} = 1.6$ eV, $t_0 = 0.27$ eV, and $t^{aa}_0 = 0.75$ eV. In order to avoid logarithmic singularities, the densities of states are averaged over the energy range $\Delta E = 7$ meV.

The main message of our work is that the mismatch between two CuO$_2$ layers generates extra van Hove singularities. For example, in Figure 3, we show the densities of states as functions of energy calculated for nine superstructures with $N_1 = 36$ and $N_2$ varying form 37 to 45. We clearly see extra van Hove singularities approximately located between two peaks corresponding to unmismatched layers. It is interesting that the number of these extra van Hove singularities exhibit a counterintuitive behavior with the changing of $N_2$: the number of peaks increases when $N_2$ decreases from $N_2 = 45$ to $N_2 = 37$. For a better comparison, we show in a single plot, Figure 4, the densities of states at the Fermi level as functions of doping calculated for all nine superstructures with different $N_2$ and fixed $N_1 = 36$.

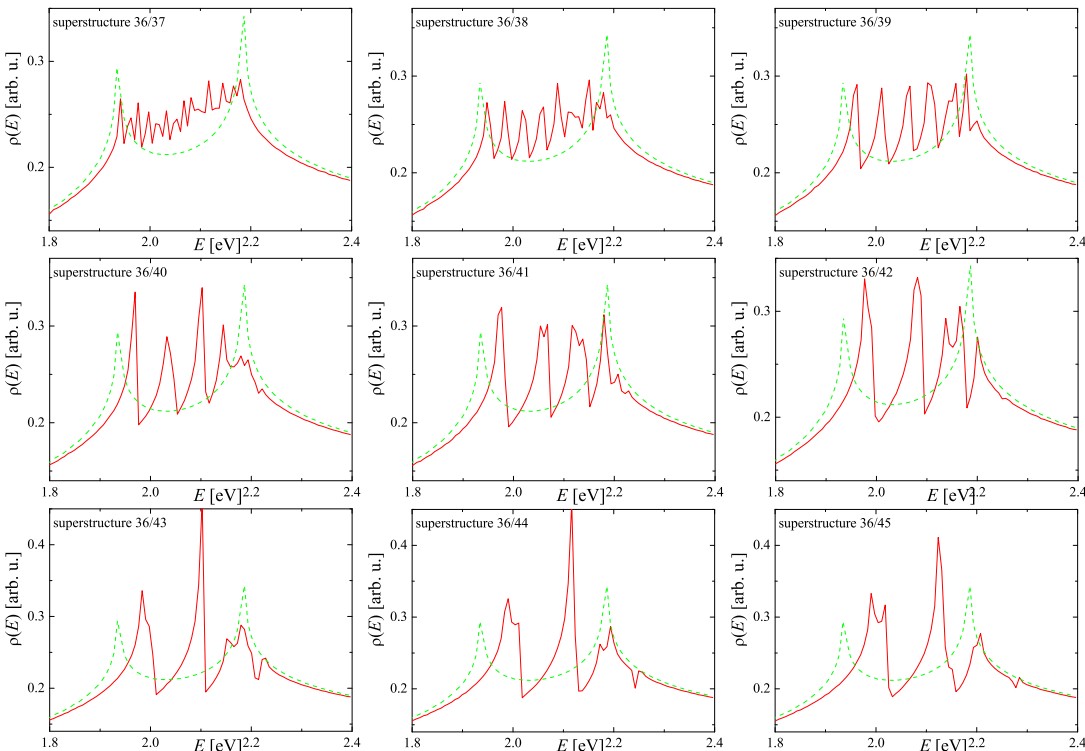

**Figure 3.** Densities of states as functions of energy calculated for nine superstructures with $N_1 = 36$ and $N_2$ varying from 37 to 45. Model parameters correspond to Figure 2 with $t_{2p} = 1.5$ eV. In order to avoid logarithmic singularities, the densities of states are averaged over the energy range $\Delta E = 7$ meV. The dashed green curve in each plot corresponds to unmismatched layers.

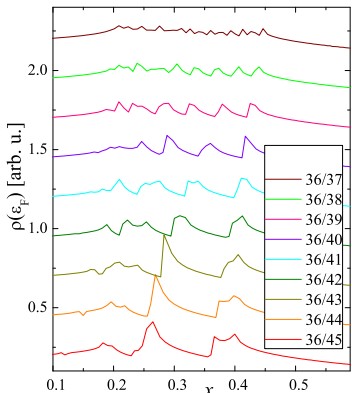

**Figure 4.** Densities of states at Fermi level as functions of doping $x$ calculated for nine superstructures with $N_1 = 36$ and $N_2$ varying form 37 to 45. Model parameters correspond to Figure 2 with $t_{2p} = 1.5$ eV. In order to avoid logarithmic singularities, the densities of states are averaged over the energy range $\Delta E = 7$ meV.

For better understanding of the origin of the extra van Hove singularities, we calculated the energy spectra for several superstructures (see Figures 5 and 6). The spectra were calculated along the triangular contour connecting the most symmetrical points $\Gamma$, $M$, and $X$ of the reduced Brillouin zone; see the right panel of Figure 1. For Figure 5, there are a lot of bands (about 100) inside the energy range shown, $1.9\,\mathrm{eV} < E < 2.25\,\mathrm{eV}$. This energy range includes two van Hove singularities of unmismatched layers (upper panel of Figure 5), located at $E_{vHs}^1 \cong 1.93\,\mathrm{eV}$ and $E_{vHs}^2 \cong 2.19\,\mathrm{eV}$. The analysis shows that the origins of these two van Hove singularities are two saddle points of certain bands located at point $M$ of the reduced Brillouin zone. In the energy range approximately between $E_{vHs}^1$ and $E_{vHs}^2$, the band crossing occurs in the region near the line connecting points $M$ and $X$ of the RBZ, as can be seen in top panel of Figure 5. Such a picture is observed for unmismatched layers. When layer 2 becomes stretched or compressed in comparison to layer 1, we can observe band flattening and band splitting in the region near the line connecting points $M$ and $X$, as is shown in middle and bottom panels in Figure 5. This band flattening gives rise to extra van Hove singularities with energies lying approximately between $E_{vHs}^1$ and $E_{vHs}^2$. The situation here is very similar to that observed in twisted bilayer graphene [2], where the band flattening is observed for large superstructures, when the twist angle $\theta$ decreases down to critical value $\theta_c \sim 1°$, and the moiré period increases as $L \propto 1/\theta$. In our case, however, the band flattening occurs only in the $k_x$-direction, since the band folding is performed only in this direction in the momentum space. As we can see in Figure 5, the bands remain dispersive in $k_y$-direction.

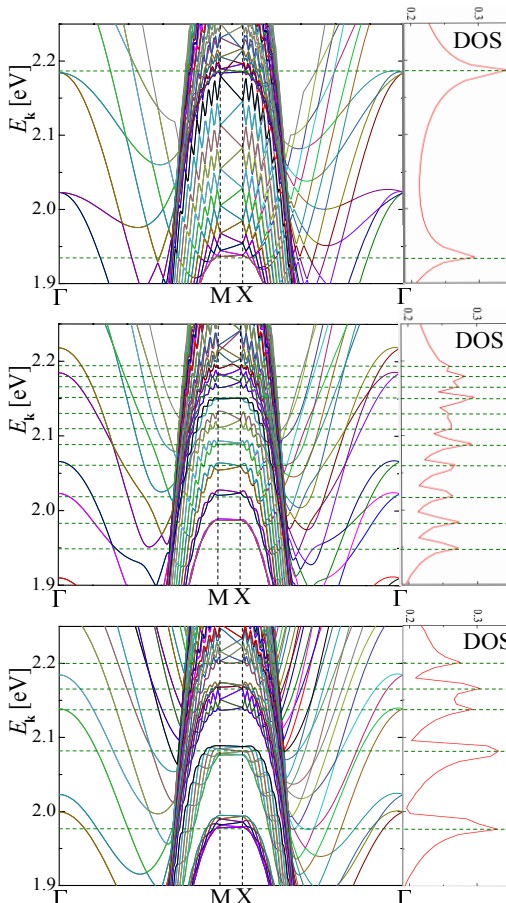

**Figure 5.** Energy spectra calculated for superstructures with $N_1 = 36$ and $N_2 = 36$ (**top** panel, no mismatch), $N_2 = 38$ (**middle** panel), and $N_2 = 42$ (**bottom** panel). The spectra were calculated along the contour shown in Figure 1. The distance between points $M$ and $X$ was enhanced by a factor of 5 for a better view. Model parameters correspond to Figure 2 with $t_{2p} = 1.5\,\mathrm{eV}$.

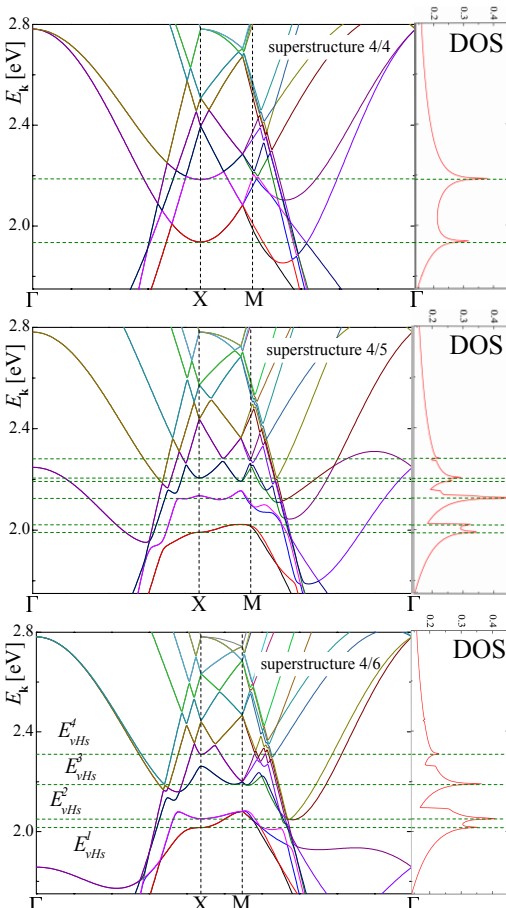

**Figure 6.** Energy spectra calculated for superstructures with $N_1 = 4$ and $N_2 = 4$ (**top** panel, no mismatch), $N_2 = 5$ (**middle** panel), and $N_2 = 6$ (**bottom** panel). The spectra were calculated along the contour shown in Figure 1. Model parameters correspond to Figure 2 with $t_{2p} = 1.5$ eV.

Note that the recent studies of 2D systems and moiré superlattices (e.g., in twisted bilayer graphene) demonstrate that such systems often exhibit higher-order VHS characterized by the power-law divergence in the density of states in contrast to the logarithmic divergence for conventional VHS [58–61]. Near such VHS, the dispersion is flatter than near conventional ones. To illustrate such a situation in our case, we show in Figure 7 a zoomed version of the highest DOS peak for the 36/43 structure. At a higher resolution, we can see that this peak splits into two. One of them has the logarithmic divergence, whereas another one exhibits the power-law divergence ($E^{-1/4}$), implying the existence of a higher-order VHS.

The spectra shown in Figure 5 contain a lot of bands. For better illustration, we calculated spectra for much smaller superlattices. Namely, we calculated spectra for superstructures with $N_1 = 4$, $N_2 = 4$, 5, and 6; see Figure 6. Since, in these cases, the superlattice cell contains much a smaller number of atoms, the energy range of interest (approximately between $E^1_{vHs}$ and $E^2_{vHs}$) contains a much smaller number of bands. For superstructures with $N_1 = 4$ and $N_2 = 4$ (no mismatch, top panel of Figure 6), we can see two parabolic bands with minima at $M$ producing two van Hove singularities at energies $E^1_{vHs}$ and $E^2_{vHs}$. By calculating the spectrum around small circle centered at $M$, we have checked that $M$ is indeed a saddle point for corresponding bands. In the momentum range near the line connecting points $M$ and $X$, we can see several band crossings. When we compress layer 2 (superstructures with $N_2 = 5$ and 6, middle and bottom panels in Figure 6), this produces the band splittings, which give rise to extra van Hove singularities.

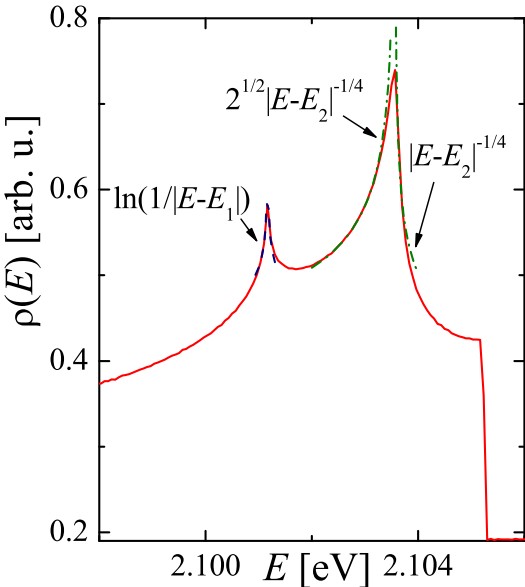

**Figure 7.** The largest DOS peak for the 36/43 structure. At a higher resolution, we can see that this peak splits into two. One of them (at $E_1$) has the logarithmic divergence, whereas the other one (at $E_2$) exhibits the power-law divergence ($E^{-1/4}$), implying the existence of a higher-order VHS. Different prefactors of the power-law divergence and the exponent of the power law agree with the predictions of [58,59]. The stepwise change in the DOS corresponds to the band edge.

## 5. Conclusions

In conclusion, we have studied the model of a double-layer BSCCO cuprate superconductor with mismatched $CuO_2$ bilayers in the diagonal direction. The mismatch between two layers produces the superstructure in the system. This gives rise to several extra van Hove singularities located approximately between energies $E_{vHs}^1$ and $E_{vHs}^2$ corresponding to two van Hove singularities of unmismatched layers. The number of these peaks and their energy positions depends on the quasi-commensurate superstructures of the *devil's staircase* in strained bilayers.

We have found that these extra van Hove singularities are closely related to the flattening and splitting of the bands inside the reduced Brillouin zone of unmismatched layers. The effect of the band flattening is similar to that observed in twisted bilayer graphene [2], where the large Fermi velocity reduction occurs for superstructures with small twist angles [62,63]. In our case, however, the band flattening appears not in the whole Brillouin zone, but only in the direction of the band folding ($k_x$-direction). In the perpendicular direction, the bands remain dispersive.

Note here that our approach relies on several approximations. For example, (i) only nearest neighbor hoppings are considered; (ii) the rotation of the orbitals in the stretched layer is not considered, and (iii) the hopping parameters in the stretched layer are assumed to remain the same as those of the non-stretched layer. Of course, these the approximations make the model rather oversimplified, and an extension of this model should indeed make it more realistic. However, just the structure of our calculations implies that the extra van Hove singularities will not disappear and certain partial band flattening should remain, but maybe in some modified manner. Indeed, the our additional calculations performed using different values of the hopping integrals in each layer lead to qualitatively similar results.

Let us also mention that DOS at the Fermi level governs to a large extent various thermodynamic properties, such as charge compressibility, spin susceptibility, and specific heat. The well-known effect of the large DOS at the Fermi level on superconductivity can be here additionally enhanced, since the large DOS may result in a strong screening of the repulsive interaction. This effect is the most clearly pronounced at the van Hove singularities, especially at strongly divergent high-order ones.

The present results show that BSCCO is an intrinsic multi-band system, where at high doping, the chemical potential is close to a Lifshitz transition appearing at VHS at the metal to superconductor transition (see also [64]). Therefore, superconductivity in BSCCO is not a single-band unconventional superconducting phase but a multi-gap superconductor with Fano–Feshbach resonance between gaps in the BCS regime and in the BCS–BEC crossover, as described by Perali et al. in cuprates [65–68] and appearing in diborides [69], oxide interfaces [70], and in room temperature hydride superconductors [71,72].

Our results are supported by the recent evidence by a high resolution STM experiment of nanoscale phase separation in a BSCCO sample doped up to 0.27 holes/Cu site [73], reaching the high doping range of the high order VHS singularities, which was predicted by theory to appear where the chemical potential approaches a high order VHS singularity near a Lifshitz transition in strongly correlated multiband systems [46–50]. The position of the Fermi level with respect to the high order VHS peaks can be controlled by the gate voltage using modern technologies. Therefore, this makes mismatched bilayer cuprates an efficient system for fine tuning of the peaks of the electron density of states with respect to the Fermi level. Note that this effect is the most clearly pronounced with rather little lattice mismatch, which produces numerous additional peaks in the density of states.

**Author Contributions:** Conceptualization, A.B.; methodology, A.B. and A.O.S.; software, A.O.S.; formal analysis, A.O.S. and K.I.K.; writing—original draft preparation, A.B., A.O.S. and K.I.K. All authors have read and agreed to the published version of the manuscript.

**Funding:** A.O.S. acknowledges the support of the Russian Science Foundation (project number 22-22-00464) for the part concerning numerical calculations. K.I.K. acknowledges the support of the Russian Science Foundation (project number 20-62-46047) for the part concerning the data analysis. A.B. acknowledges the support of the Ministry of Science and Higher Education of the Russian Federation (Agreement No 075-15-2021-1352).

**Institutional Review Board Statement:** Not applicable.

**Informed Consent Statement:** Not applicable.

**Data Availability Statement:** Not applicable.

**Acknowledgments:** We are grateful to the Joint Supercomputer Center of the Russian Academy of Sciences (JSCC RAS) for the provided computational resources.

**Conflicts of Interest:** The authors declare no conflict of interest.

## Abbreviations

The following abbreviations are used in this manuscript:

| | |
|---|---|
| DOS | density of states |
| VHS | van Hove singularities |
| QCP | quasi-commensurate phase |
| BSCCO | $Bi_2Sr_2CaCu_2O_{8+y}$ |
| BSCYCO | $Bi_2Sr_2Ca_{1-x}Y_xCu_2O_{8+y}$ |
| BZ | Brilouin zone |
| CDW | charge density wave |
| HEPs | high entropy perovskites |
| STM | scanning tunnel microscopy |
| BCS | Bardeen–Cooper–Schrieffer |
| BEC | Bose–Einstein condensate |

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
