# Peer review of "Moiré-like Superlattice Generated van Hove Singularities in a Strained CuO2 Double Layer"

_condensedmatter, doi:10.3390/condmat7030050_

Round 1

Reviewer 1 Report

The authors have investigated an useful model of double layer BSCCO cuprate  system with mismatched CuO2 bilayers in diagonal direction. The mismatch gives rise to several van Hove singularities. Several consequences are presented in this work.

I recommend this paper for publication.  

A suggestion is given to check English of the paper. Many long sentences may be made a few short sentences.

A word ‘the’ appears unnecessary in the abstract in line 6 and in the Introduction -line 77.

Author Response

We are grateful to Referee 1 for the positive attitude to our work and useful comments.
According to the recommendation of Referee 1, we divided several sentences into shorter ones (see lines 27-29, 36-37, 64-65, 71-73, 109-110, and 288-289). We also corrected the typos related to unnecessary 'the'.

Author Response

We are grateful to Referee 2 for the positive attitude to our work.

Reviewer 3 Report

Report on condensedmatter-1865904

==========================

The paper presents a new idea. I recommend

acceptance in the present form.

Author Response

We are grateful to Referee 3 for the positive attitude to our work.